# Dietary Approaches to Iron Deficiency Prevention in Childhood—A Critical Public Health Issue

**DOI:** 10.3390/nu14081604

**Published:** 2022-04-12

**Authors:** Jean-Pierre Chouraqui

**Affiliations:** Pediatric Nutrition and Gastroenterology Unit, Woman, Mother and Child Department, University Hospital of Lausanne, 1011 Lausanne, Switzerland; chouraquijp@wanadoo.fr; Tel.: +33-608-276-543

**Keywords:** infants, children, anemia, iron intake, biomarkers, iron rich foods, iron-fortified foods, meat, cow’s milk, formula

## Abstract

Iron is an essential nutrient, and individual iron status is determined by the regulation of iron absorption, which is driven by iron requirements. Iron deficiency (ID) disproportionately affects infants, children, and adolescents, particularly those who live in areas with unfavorable socioeconomic conditions. The main reason for this is that diet provides insufficient bioavailable iron to meet their needs. The consequences of ID include poor immune function and response to vaccination, and moderate ID anemia is associated with depressed neurodevelopment and impaired cognitive and academic performances. The persistently high prevalence of ID worldwide leads to the need for effective measures of ID prevention. The main strategies include the dietary diversification of foods with more bioavailable iron and/or the use of iron-fortified staple foods such as formula or cereals. However, this strategy may be limited due to its cost, especially in low-income countries where biofortification is a promising approach. Another option is iron supplementation. In terms of health policy, the choice between mass and targeted ID prevention depends on local conditions. In any case, this remains a critical public health issue in many countries that must be taken into consideration, especially in children under 5 years of age.

## 1. Introduction

Iron is an essential nutrient for all living organisms, especially due to its importance in ensuring the oxygen-carrying capacity of hemoglobin (Hb) and its key function in tissue oxygenation, as well as in other biochemical functions, such as DNA synthesis and enzyme activities [1,2]. Most data on iron homeostasis have been established in adults and were recently reviewed [1,2,3,4,5,6,7]. They are schematically described in Figure 1.

In the absence of a pathway for iron excretion, iron homeostasis is determined by the regulation of iron absorption and traffic in the body, which are regulated by iron-regulatory proteins [2,6]. Iron absorption depends on the body’s iron status, the rate of erythropoiesis, the amount and form of iron in the diet, heme iron (Fe^2+^) or inorganic iron (Fe^3+^), and the presence of absorption enhancers and inhibitors in the diet [1,5,6,8,9,10]. Hepcidin, a liver hormone, is the master regulator that coordinates absorption, storage, and macrophage release [5]. Hepcidin secretion is increased by transferrin saturation and enhanced liver iron stores, as well as inflammation, while hepcidin suppression upregulates iron absorption and recycling in the case of iron deficiency (ID) or increased erythropoiesis hepcidin suppression [1,2,5,11]. Young infants seem to have a lower capacity to regulate iron homeostasis [12].

Depletion in body iron stores may lead, when combined with an inadequate intake of bioavailable iron, to ID, which will precede the onset of iron deficiency anemia (IDA) [11]. ID is the most common nutritional deficiency in children who are at particular risk for developing it, due to their rapid growth and the use of complementary foods with low bioavailable iron content. ID prevalence in children remains high in many countries, and ID is the world’s top-ranking cause of anemia, although its importance varies by region and socioeconomic status [13]. In addition, several risk factors have been identified.

Given the implications of ID and IDA for the clinical health of individuals and broader implications for public health, a prevention strategy is an important public health issue.

The importance of an appropriate dietary iron intake (DII) in this prevention was highlighted by the recent evidence of a positive correlation of iron intake in children with ferritin (*r* = 0.308, *p* = 0.002) and Hb level (*r* = 0.769, *p* < 0.001) [14]. Considering, therefore, the hypothesis that iron DII may prevent the depletion of iron stores, after first briefly recalling the ID problem, the aim of this narrative review was (i) to review information on recommended intakes in children, (ii) to summarize current knowledge of iron dietary intakes in children and relate that information to recommended intakes, (iii) and then to identify potential strategies to address suboptimal intake by describing common food sources of iron in childhood and different methods of food fortification. To the best of our knowledge, no recent review has addressed all these considerations.

## 2. Iron Deficiency

### 2.1. Etiology

Depletion in iron stores leads to ID as a consequence of a negative iron balance, which may be due to the interplay of increased iron requirements, limited external supply and/or absorption, and blood loss. The risk factors listed in Table 1 can be summarized in five categories:Inadequate DII, which is reviewed below, is related to prolonged exclusive breastfeeding in infants, or the insufficient intake of iron-rich foods considering the growth velocity at any age or menstruation in adolescent girls;Low iron stores in the neonatal period may be due to a short gestation duration in the case of a preterm birth or a low birthweight, a maternal IDA, or an early cord clamping [15,16];Malabsorption including celiac disease [17] and parasitic infections [18,19];Blood loss [19];Low socioeconomic status [13,20,21,22,23,24], whereby economic growth was shown to be associated with reductions in anemia among school-aged children [21,24].

**Table 1 nutrients-14-01604-t001:** Factors that may negatively influence iron balance according to childhood stage.

Neonatal Period	Infancy	Childhood and Adolescence
Maternal iron deficiency anemia	Low neonatal iron stores	Low dietary iron supply and/or bioavailability
Fetal–maternal hemorrhage	Rapid growth	Vegetarian or vegan diet
Twin-to-twin transfusion	Frequent blood sampling	Pica; pagophagia
Premature birth	Prolonged exclusive breast feeding	Intense physical activity
Low birth weight	Cow’s milk feeding	Antacid therapy
Early umbilical cord clamping	Low-iron-content complementary diet	Esophagitis
Phlebotomy losses	Cow’s milk allergy	*Helicobacter pylori* infection
Esophagitis	Intestinal malabsorption (Celiac disease, short-bowel syndrome)	Intestinal malabsorption (Celiac disease, short-bowel syndrome)
Intestinal blood loss	Antacid therapy	Parasitosis (Giardia, hookworm)
Erythropoietin administration	Esophagitis	Use of nonsteroidal anti-inflammatory drugs
	Lead exposure	Lead exposure
	Intestinal blood loss	Gastrointestinal blood loss (gastritis, varices, Meckel’s diverticulum, ulcerative colitis, vascular malformations, tumors, polyp)
		Obesity
		Excessive menstrual losses
		Adolescent pregnancy
	Low socioeconomic status	Low socioeconomic status

### 2.2. Development of Iron Deficiency

ID may progress through three stages from an iron depletion to iron-deficient erythropoiesis and to the most severe stage of IDA [10,12]. These different stages are characterized by the presence or absence of clinical, biochemical, hematological, and functional signs (Table 2) [19].

### 2.3. Epidemiology

According to the Global Burden of Disease Study 2017, ID had, for all ages and both genders combined, a prevalence of over 1.1 billion individuals worldwide and was shown to be one of the five leading causes of years lived with disability [25]. In 2019, the worldwide prevalence of anemia was approximately 40% in children <5 years and approximately 60% in Africa [20]. ID accounted for 42% of anemia burden from 1990 to 2010, mainly in Central and West Africa, in South Asia, and in socioeconomically disadvantaged areas [13]. In South Africa, the anemia prevalence from 1997 to 2021 was estimated at approximately 61% among children under five, with the highest values in children aged 1–30 months and the prevalence of ID and IDA at 10.7% and 28.9%, respectively [26]. From 1995 to 2011, thanks to health policy and socioeconomic improvement in some areas, global anemia prevalence decreased by 4–5 percentage points [20,27], but the youngest age groups had the least favorable changes from 1990 to 2010 [13]. However, from 2010 to 2019, the anemia prevalence in children under five was stagnant [20].

Using random-effects meta-analysis techniques on data from 23 countries for preschool children, the global prevalence of ID and IDA was 17.3% (95% CI: 13.8, 20.8) and 9.6% (95% CI: 7.2, 12.0) [28]. The proportion of IDA in anemia was estimated to be 24.6% (95% CI: 18.0, 32.0), which is lower than that assumed by others [28]. This proportion is even lower as it concerns countries with a high prevalence of global anemia or with high inflammation exposure [28]. A lower prevalence was observed in central Asia (11.0%) and the highest in the Middle East (35.6%), while it was 24% in southeast Asia, 26.9% in Latin America, and 28.1% in sub-Saharan Africa. Large epidemiological differences, including adolescents, are indeed observed between countries, [22,23,29,30,31,32,33,34,35,36,37] and ethnic groups [38].

### 2.4. Health Consequences of Iron Deficiency

The main public health problems associated with ID in childhood are the progression to anemia and the risk of poor neurodevelopment, the latter being essentially related to the former, aside from immediate signs of ID that are dependent on severity [3,6,10,15,39,40,41,42,43].

ID limits adaptive immunity and responses to vaccination [44,45]. However, the relationship between ID and the risk of infection remains unclear [6,10,15].

While a negative association could be shown between growth velocity and serum ferritin levels over the age of 2–6 months among healthy infants [46], the association between ID and IDA and poor linear growth remains controversial, and nutrition-based interventions containing iron have no significant effect on growth [47,48]. The association between ID and obesity remains unclear in terms of the sense of the cause–effect relationship [49,50].

## 3. Iron Requirements

### 3.1. Requirements According to Age Groups

The requirements for absorbed iron in children aim to replace the obligatory losses and to ensure the synthesis of new tissues involved in their growth rate, including the increase in Hb mass. Iron requirements are particularly high during the two critical periods of rapid growth, namely, early childhood and adolescence [3,51]. During the first years of life, iron is also critical for the development of the central nervous system, since it is involved in brain energy metabolism, myelination, and neurotransmission [39].

#### 3.1.1. Infancy (Birth through to 11 Completed Months of Age)

A term newborn infant has approximately 75 mg of iron per kg of body weight, 60% of which is accreted during the third trimester of gestation with an increase in ferritin concentration [52,53]. Maternal ID, with or without IDA, as well as maternal diabetes mellitus, may adversely affect fetal iron status [6,52]. However, a recent meta-analysis found an inconsistent association between maternal and child iron status [54]. Newborn iron is distributed as 75–80% in erythrocytes as fetal Hb, the concentration of which is high at birth, as 10% in tissues such as muscle myoglobin, and as 10–15% in stores mostly as ferritin, especially in the liver [52]. The liver is one of the main iron storage organs in fetal life, but the relationship between hepatic iron concentration and gestational age is controversial [55,56].

Factors that may influence the body iron status in the perinatal period are listed in Table 1. Low-birthweight and preterm infants have lower iron stores at birth and higher requirements due to more rapid postnatal growth and increased losses as a result of frequent iatrogenic phlebotomy [16]. During the first 2 months of life, the production of fetal Hb is replaced by that of adult Hb, leading to a decline in Hb concentration, while adult-type erythrocytes have a longer life span (120 days vs. 60–90 days for fetal erythrocytes) [57]. Subsequently, iron stores become augmented, and the ferritin concentration increases. During the following months, iron is transferred back from stores to the blood compartment, making the term infant virtually self-sufficient in iron during the first months of life [10]. Hence, the requirement for dietary iron during this period is minimal but depends greatly on the iron stores at birth.

By 4–6 months, iron stores decrease significantly, and the infant needs a generous DII while they triple their body weight and double their iron stores. The requirement of absorbed iron in infants aged 7–11 months was estimated to be approximately 0.8 mg/day [3].

#### 3.1.2. Toddlers (1–3 Years of Age) and Children

Toddlers’ iron needs continue to be relatively large and are not always met by the typical toddler’s diet. Then, iron requirements increase slightly, owing to the slowdown of growth velocity and the decrease in endogenous losses [3]. The body iron related to body weight is relatively stable at approximately 40 mg/kg, while the average weight gain is approximately 7 kg between 1 and 4 years of age, 7.4 kg from 4 to 7 years of age, and 18 kg between 7 and 12 years of age [3]. The requirement for absorbed iron was estimated at 0.5 mg/day in children aged 1–6 years and at approximately 0.8 mg/day in children aged 7–11 years [3].

#### 3.1.3. Adolescents (12–17 Years)

With puberty, iron requirements increase in both genders as a result of the acceleration of growth, along with an increase in lean body mass and an expansion of the total blood volume, to which the compensation of the periodic menstrual loss in girls after menarche must be added [58]. Of note is the wide variation of age at menarche. Between the age of 12 and 18 years, the average weight gain of boys is 27.8 kg and that of girls is 14.8 kg, and their requirements for absorbed iron are 1.27 and 1.13 mg/day, respectively [3].

### 3.2. Dietary Reference Values for Iron Intake

The dietary reference values (DRVs) set by different authorities and used to evaluate intake are very different from one another (Table 3). For infants up to 6 months, iron DRVs are estimated as adequate intake (AI), according to the average iron supply from breast milk [59,60]. For older infants and children, a factorial modeling is used, providing estimated average requirements (EAR) or population reference intake (PRI) [3,59,61]. The explanations for the differences in values include different bases for calculation and different estimations of dietary iron bioavailability. The EAR is the calculated daily intake level of dietary iron sufficient to meet the needs 50% of healthy children; the PRI is the adequate level for at least 97.5% of children and is sometimes given as the recommended dietary allowance. It is quite difficult to currently assess the adequacy of DII based on the EAR cut-point method, which is considered to underestimate the true prevalence of inadequacy [3,62]. Probabilistic methods, which consider both intake and requirement variability, might be a useful alternative provided that the iron requirement distribution is determined, which has only been carried out by the IOM [59,62]. Moreover, the DRVs do not fully account for the differences in the relative bioavailability of iron.

## 4. Actual Dietary Iron Intakes in Children

DII determination is considered as the only available marker to assess the risk of ID [6]. The DII is estimated from food consumption surveys. The results are difficult to compare from one country to another given the differences in the dietary assessment methods, tables of food composition used, population studied in different environmental and socioeconomic conditions, and reference values used to assess the adequacy of intakes. All surveys showed intakes more or less differing from recommendations according to countries and mode of feeding and a great interindividual variation whatever the age (Table 4) [3,22,58,63,64,65,66,67,68,69]. Data from other countries are scarce or are limited to specific areas in the country and/or special populations.

Most of the average DIIs reported are far lower than the reference values, and, even if their inadequacy cannot be asserted, given the reservations expressed above, they should be considered from the perspective of the risk of ID. However, according to their design and population enrolled, studies showed either a correlation between DII and iron status [14,70] or no correlation [71,72]. Nevertheless, monitoring the DII of children is necessary in each country to determine whether they are meeting the recommendations, and to guide public health strategies [6].

## 5. Dietary Iron

Given that ID prevalence in children remains high in many countries, and considering the subsequent health consequences, ID prevention is of global public health concern. The potential strategies to address optimal intake appeal first to offer a diverse diet with a variety of iron-rich foods with high iron bioavailability, alongside other strategies to reduce the risk factors of ID. A low iron content in the diet is indeed, apart from special risk factors or possible iron losses, the main reason for suboptimal intake, while not forgetting the importance of iron bioavailability.

### 5.1. Dietary Iron Bioavailability

The impact of DII on the body iron status is hampered by the iron bioavailability, i.e., absorption, which is determined by the body iron status, the chemical form of iron, and the co-ingestion of enhancers and inhibitors [1,2,6,8,9,10]. Heme iron, whose absorption is estimated to be 15–35%, is provided by meat. Nonheme iron, whose absorption ranges between 5% and 15%, and which includes ferritin iron, may be provided by both animal and plant food sources [5,6].

The importance of the effect of the diet composition on iron bioavailability is somewhat controversial, and this bioavailability ranges between 0.7% and 23%, with a generally mean accepted value of 10% or even lower in people with normal iron stores [2,3,9,10]. One obviously cannot reason only by using a single food approach but by considering the whole meal, especially the balance between enhancing (ascorbic acid, heme iron, and fermented vegetables) and inhibiting factors (mainly phytate and phenolic compounds, such as in tea or coffee, but also calcium, zinc, copper) [2,6]. The iron content of the diet must then be taken into account by integrating these concepts.

### 5.2. Dietary Iron Sources

#### 5.2.1. Dietary Recommendations and Usual Feeding Pattern in Children

Feeding children aims to meet all nutritional needs while accompanying the child in making the gradual transition from a milk diet to a predominantly solid food and adult type food.

In early life, according to the WHO, infants should be exclusively breastfed up to 6 months and partially breastfed together with appropriate complementary feeding up to 2 years [73]. This recommendation is advocated by most African and Asian countries but also some Western countries, such as Canada [74]. In some other areas, such as the USA [75] or the European Union [76,77], authorities recommend exclusive breastfeeding for a minimum of 4 months. In fact, the average prevalence of exclusive breastfeeding at 0–5 months is approximately 47% in low-income countries, 39% in lower–middle-income countries, and 37% in upper–middle-income countries with important differences between countries within the same group [78]. A strong inverse correlation (Pearson’s *r* = −0.84; *p* < 0.0001) is noted between breastfeeding at 6 months and log gross domestic product per person [78]. Non-breastfed infants or partially breastfed infants are usually formula-fed, i.e., a milk-based manufactured breast milk substitute designed to satisfy, by itself, the nutritional requirements of infants during the first months of life up to the introduction of appropriate complementary feeding, as defined by the FAO–WHO Codex Alimentarius [79]. After 6 months of age, according to countries and cultural habits, infants who are weaned off breast milk consume more or less formula. The prevalence of continued breast feeding at 12 months is approximately 94% in low-income countries and 25% in high-income countries [78]. It is the highest in sub-Saharan Africa, south Asia, and parts of Latin America, and it can be as low as less than 1% in some Western countries, such as the UK [78]. The WHO recommends starting complementary feeding at 6 months [73]. In fact, the recommended and actual timing of complementary feeding varies according to countries, cultural habits, food availability, and socioeconomic status, but is on average between 4 and 6 months. This period corresponds precisely to the time when endogenous iron stores have been used up and the need for exogenous iron increases. The earlier age of introduction of complementary food is mainly noted in Western industrialized countries [80], whereas it is between 6 and 8 months in low- and middle-income countries [81]. Gradually, with age and according to cultural habits, children will consume increasingly like adults and will participate in family meals.

#### 5.2.2. Breastmilk

Since the iron requirement is minimal during the first months of life of normal-birthweight infants with sufficient stores at birth and without risk factors of ID, breast milk is sufficient to supply this need, at least up to 4 months, despite its low iron content (~0.2–0.5 mg/L), regardless of the mother’s intake [10,82,83,84,85]. Iron absorption from breast milk is usually assumed to be up to 50% [86], but a later study found a bioavailability of 16.4% at 6 months [87]. From its higher level in colostrum milk, iron concentration declines throughout the first few months of lactation [85], as well as the infant’s plasma ferritin level [88]. Given this and the concern regarding the decrease in iron stores over several months, controversy arose regarding the iron needs of breastfed infants after the age of 4 months, especially in infants with low iron stores at birth (low-birthweight infants, infants from diabetic mothers, early cord clamping, or with a low weight gain since birth) [10,15,16,82,89,90,91]. Exclusive breastfeeding beyond 4 months was indeed associated with increased risk of ID [10,22,31,35,48,67,88]. Complementary food rich in iron or medicinal iron supplementation improves the iron status of breastfed infants [84,88,92,93]. Thus, the American Academy of Pediatrics recommended that exclusively breastfed term infants receive an iron supplementation of 1 mg/kg per day, beginning at 4 months of age and continued until appropriate iron-containing complementary foods have been introduced [82]. However, the ESPGHAN committee on nutrition questioned these recommendations except for in high-risk infants (low socioeconomic status or high prevalence of IDA), arguing for insufficient evidence in healthy European children [10]. Recent reported data are in favor of a necessary supplementation [65,94]. At present, the question is, therefore, far from settled [91]. The association between the duration of exclusive breastfeeding beyond 6 months and the occurrence of ID has been documented [48,95,96,97]. The odds of ID were shown to increase by 4.8% (95% CI: 2–8%) for each additional month of breastfeeding [95].

#### 5.2.3. Formula

The importance of iron fortification of children formula for the prevention of ID was demonstrated for over 60 years for infant formula, over 30 years for follow-on formula, and more recently for young child formula [10,22,23,34]. The absorption rate of iron from formula was estimated to be approximately 10% [11]. In most surveys, formula was the main source of iron up to 2 years of age in non-breastfed children [64,67,68,98,99]. The prolonged use of iron-fortified formula up to 3 years was highlighted by several studies and is correlated with the DII [22,100,101,102,103,104,105]. This is especially important as the consumption of iron-rich complementary food is low, with a poor diet diversity. However, attention has been drawn to possible deleterious effects in infants without ID risk fed formula with a high concentration of iron. Such feeding may result in poor copper but not zinc absorption [106,107]. Infants with high levels of Hb (>12.8 g/dL) who were fed a high-iron-fortified formula (12.7 mg/L) between 6 and 12 months were shown to have lower cognitive and visual motor scores at 10 and 16 years than those who received low-iron formula (2.3 mg/L) [108,109]. The last recommendation of the American Academy of Pediatrics dates from 2010 and was 10–12 mg/L [82]. In 2014, The ESPGHAN considered that infant formula (up to 6 months of age) should have an iron content of 4 to 8 mg/L [10], while the EFSA proposed a content of 0.3–1.3 mg/100 kcal in infant formula (approximately 2.1–9.1 mg/L) and 0.6–2 mg/100 kcal (approximately 4.2–14 mg/L) in follow-on formula [110]. A recent RCT conducted in healthy term infants confirmed that the fortification of infant formula with 2 mg/L of iron was adequate [111].

#### 5.2.4. Cow’s Milk

Cow’s milk has a low iron content (0.3–0.6 mg/L) with low bioavailability, which is impaired by the high content of casein and calcium [112,113]. Moreover, the ingestion of cow’s milk may cause occult blood loss in feces, which may affect up to 40% of otherwise healthy infants and younger toddlers [113]. Blood loss may be, on average, 1.7 mL/day, which is equivalent to iron loss of 0.53 mg/day. There is extensive evidence showing the deleterious effect of early cow’s milk consumption above 400 mL/day by infants on iron status and on the increased probability of ID [27,28,29,30,95,96,113]. Each month of cow’s milk consumption increased the risk ID and IDA by 39% and 18%, respectively [30]. Cow’s milk should not be used as the main drink before 12 months of age [76,82]. Clearly, neither infants nor toddlers can depend on cow’s milk to meet their iron needs; instead, they depend on fortified formula and iron-rich foods or iron supplements.

#### 5.2.5. Usual Solid Foods

Measures to prevent ID after the age of 4–6 months include a sufficient intake of iron-rich complementary foods (Table 5), but most complementary foods are low in iron unless they are fortified with iron.

##### Limitations in Iron-Rich Food Consumption

The benefit of the iron-rich food may be limited by the iron bioavailability and the quantity consumed. With age, this quantity increases, as well as the weight of the recommended portion size. Portion size may be determined by the weight or volume of household utensils such as tablespoons, hand measures, cups, or reference objects whose sizes vary by country [116,117,118]. Portion size itself and food choice also vary widely between countries, across different population groups, and according to the cultural environment, parents’ dietary habits, and the child’s own choice [116,117,118,119,120]. Hence, the professional will have to adapt recommendations to each child in each environment by determining the appropriate food portion, described in practical means, according to the theoretical iron content and estimated bioavailability. They should take into account the usual family meals as a whole, as well as the overall daily diet.

Iron-rich foods include animal tissues, mainly red meat, offal, shellfish, eggs, pulses, and nuts. Vegetables and fruits have a low content of iron, except for dried fruits.

##### Sources of Heme Iron

Owing to their heme iron content, which is higher for beef than pork and poultry, meats are highly valued sources of iron, with an absorption of approximatively 25%, which is not affected by dietary factors, except calcium [10,76,121,122,123]. Moreover, the presence of heme iron in a meal enhances the nonheme iron absorption from foods consumed at the same time via an unclear mechanism. In young children, the association between red meat consumption and iron status has rarely been found to be significant [10,99,124], except in cases of a high-meat diet [10]. Red meat, however, helps to prevent ID [10,99,124,125], but it is only sparingly consumed by infants and young children [64,67,126]. Older children and adolescents are more likely to consume meat products [127]. Income per capita, natural endowment factors, meat prices, and culture are major worldwide drivers of red meat consumption [128]. Special attention must be paid to consumers who follow the Kosher or Halal rules, especially in the context of low socioeconomic status, as these populations have been shown to be more likely to develop ID [129]. This is especially the case for the most observant believers and for those who, for practical reasons, deviate from the rules by decreasing their meat consumption. On the other hand, processed and cooked meat when boiled, braised, or roasted has decreased heme iron content, with an average loss of 17% from the first few minutes of heating at 60 °C and of 50% after 60 min at 95 °C [3,122,130]. The use of iron-containing cookware or a fish-shaped iron ingot could serve as a means of reducing ID and IDA [131,132,133]. Iron leaching and absorption, therefore, depend on the type of food prepared, especially food acidity, mainly including ascorbic acid [131,132,133,134,135,136], whereas they are inhibited by tannic acid and water contamination with arsenic or manganese [133,136]. It should be considered that, despite its significant contribution to iron intake, excessive meat intake may have negative health consequences due to the overconsumption of energy and fat [137].

##### Nonheme Iron

Inorganic iron, which is mainly found in products of plant origin, accounts for more than 80% of the iron in a standard diet, but its bioavailability is low (1–12%) [3,57,138,139,140]. Facilitators of its absorption include meat, as well as ascorbic and citric acid, which may counterbalance the effect of inhibitors such as calcium, casein, dietary fiber, phytates, and polyphenols such as in tea or coffee [138]. The result is that the individual effects of dietary inhibitor factors may be reduced when they are consumed as part of a whole diet, suggesting that the overall effect of enhancers and inhibitors on iron absorption is considerably less than predicted from single meal studies [138].

#### 5.2.6. Fortified Foods

##### Addition of Iron to Staple Foods

Several studies have shown that, especially in low-income countries, iron food fortification, referring to the addition of iron alone or with other micronutrients during food processing, was a safe and cost-effective way of preventing ID despite some technical difficulties related to undesirable changes in the food, such as alterations in appearance and taste [10,82,141,142,143,144,145,146]. Thus, various commonly consumed foods have been fortified in addition to formula, mainly cereals [10,82,92,93,142,147,148], but also wheat flour [143,145,146,149,150], maize flour [151], rice [143,146,152,153], soy sauce or fish sauce [146,154,155,156], salt [157,158], and candies [159]. Such strategies may, however, have a limited impact in infants and young toddlers due to their relatively low staple food consumption and even lower seasoning and condiment intake [160]. In at least 85 non-European countries, cereals or flours have been mandatorily fortified with iron [161]. Fortified cereals and other products have been shown to be effective in the prevention of ID in young children [10,82,92,93,147,148]. Conversely, the meta-analysis performed by Eichler et al., including 24 RCTs with iron-fortified cereals or dairy products, showed only a marginal health effect in children 5–15 years [142]. Several biases were, however, acknowledged by the authors, mainly linked to the fact that most of the studies were performed in malaria-endemic zones and/or in areas with a low prevalence of ID. Results with fortified flour are even more inconsistent [141,145,146,149,150].

##### Biofortification

Biofortification consisting of the breeding or genetic modifications of staple crops in order to select those with a higher iron and/or lower phytate content has emerged as a promising, feasible, and cost-effective approach to prevent ID [141,162,163,164]. Such strategies have been applied to several plant species, mainly beans [164,165,166,167,168], but also rice [165,166], pearl millet [164,165,166,169], and cowpea [170]. They have been mandated by several developing countries [165]. Their effectiveness has some limitations related, on the one hand, to the small quantities of these foods sometimes consumed, especially by young children and, on the other hand, to the large differences in iron content according to the numerous plant varieties and to the effective iron bioavailability [2,166,167,168,171].

## 6. Discussion and Strategies

### 6.1. Limitations

Even if few studies have been carried out in children, it can be considered that, as in adults, the iron status of an individual is determined by the regulation of iron absorption in the proximal small intestine. Iron absorption increases in the case of ID, hypoxia, or accelerated erythropoiesis, and decreases in the case of iron overload and inflammation, thanks to hepcidin regulation [12]. It should also be noted that, especially in preterm infants, no relationship was shown between iron absorption and iron intake or ferritin, but there was a correlation with transferrin saturation [172].

Data on ID prevalence from some countries are obviously lacking and needed for different age groups. Nevertheless, considering the level of ID prevalence, which remains high in many countries, and in light of the accumulative evidence regarding the adverse effects of ID on children’s outcomes, a comprehensive prevention approach, particularly among children under 5 years of age, is justified in terms of health policy [3,10,51,82,173]. This prevention is obviously all the more important as ID prevalence is high. Conversely, as discussed above, concerns must be expressed regarding the use of highly iron fortified food in children without ID risk [106,107,108,109].

There is, therefore, a need for sufficiently powered randomized controlled studies of the effects of different levels of iron fortification to better establish the appropriate dose. Such trials should assess the effects of iron fortification not only on iron status but also on growth, neurodevelopment, and health outcomes. In particular, whether preventive measures prevent brain dysfunction induced by ID is yet to be determined. Two reasons led us to consider with caution the conflicting data on the relationship between systemic markers of iron status and iron intake. First, the results depend on the ID prevalence in the studied population. Second, the efficiency of increasing mean iron DII is principally influenced by iron absorption, which is driven by the systemic iron requirements. All the limits outlined above must be considered to define the most appropriate strategy according to local conditions.

### 6.2. Strategies to Prevent Iron Deficiency 

#### 6.2.1. Dietary Strategies to Address Suboptimal Intake

##### In Infants and Toddlers

Breastfeeding should be encouraged [74,75,76]. However, when exclusive breastfeeding is extended beyond the age of 4 months, the advisability of supplementation should be discussed according to the clinical and socioeconomic context [10,22,31,35,48,67,82,84,88,89,90,91,92,93,94,95,96,97]. Otherwise, an iron-enriched formula must be chosen and used after 1 year and if possible, for up to 3 years [10,22,23,34,64,67,98,100,101,102,103,104,105]. When solid foods are introduced, the prevention aims primarily to improve the quality of the diet, by favoring natural bioavailable iron-containing foods, especially red meat, poultry, or fish (Table 5), and food rich in vitamin C [10,76,121,122,123]. The complementary options are to promote the supply of iron-fortified foods [10,82,92,141,142,143,144,145,146,147,148,149,150,151,152,153,154,155,156,157,158,159] or biofortified foods [139,162,163,164,165,166,167,168,169,170] and to delay the introduction of cow’s milk at least until after the age of 1 year or even after 2 years [76,82].

##### In Children over 3 Years Old and Adolescents

There are no specific recommendations regarding the dietary prevention of ID in these age groups. Common sense would lead to suggest the same proposals as for the youngest age groups, namely, the consumption of iron-rich food (Table 5), such as red meat, while avoiding excess intake, and the consumption of citrus fruits [56]. With age, the diet becomes the same as that of adults, and the dietary recommendations correspond to the general recommendations, as formulated by the WHO, and fortified or biofortified foods may be introduced [141]. Prevention in menstruating teenage girls is similar to that proposed for young women, which reinforces the general recommendations [174]. Children and adolescents should also avoid drinking tea or coffee with a meal, due to their inhibitory effects on iron absorption [15,141,175]. In fact, in areas with a high ID prevalence and in at-risk populations, prevention is mainly based on iron supplementation [51].

##### Cost/Benefit Ratio

The obvious limitation to a dietary approach for preventing ID is the cost of iron-rich foods, especially in unfavorable socioeconomic conditions. That is why such a prevention can only be integrated into a public health policy and the education of populations and healthcare providers. Solutions exist to enrich staple foods such as cereals or milk with iron at a relatively low cost or, even better, to promote staple crop biofortification [141,143,144,162,164]. Both strategies are regarded as cost-effective (the cost to achieve ID prevention) and as having the potential to achieve a high cost/benefit ratio (the cost of the intervention compared to the cost of ID) [164].

#### 6.2.2. Choice of a Health Policy

##### Implementation of Dietary Measures

The question that may arise is that of the choice between a mass fortification, which will be regulated by governments, or a targeted fortification directed toward high-risk groups, which is more complicated to implement. This choice must consider local conditions and the level of ID prevalence [141,144]. Recommendations mainly focus on infants and toddlers. Everywhere, in addition to actions to improve socioeconomic status, hygiene, and sanitation conditions [141], and to delay cord-clamping at birth [15,176,177], the consumption of a balanced diet including iron-rich foods is recommended for all children [10,76,82,141]. Promoting increased meat consumption is not the most appropriate solution in low-income countries or in countries where meat or protein intake is already high, even if it can contribute to increasing iron intake. Moreover, some caution must be observed as to the amount of iron supplied in non-iron-deficient children, especially regarding infant formula with a high iron content [108,109].

Special attention must be paid to premature or low-birth-weight infants, to children with a low socioeconomic status and/or with an obvious low intake of iron-fortified products, including formula and/or low meat intake, and to those with a high cow’s milk intake (introduced before 1 year of age or intake >400 mL afterward). These children, in particular, may need to consume iron-fortified foods or even supplementation [10,82,141,177]. Preterm and low-birth-weight infants should be given 2 mg/kg/day from 6 weeks to 6–23 months of age [16,141].

Healthcare providers must also be aware of vegetarian and especially vegan diets and provide nutritional education on their potential risk, together with practical feeding recommendations [178,179].

Mass prevention is aimed at children from countries with a high prevalence of ID (>10%) or IDA (>5%) [141,177,180]. According to local conditions, interventions can then use the iron fortification of commonly consumed foods [10,82,92,93,141,142,143,144,145,146,147,148,149,150,151,152,153] or the biofortification of staple crops [141,162,163,164,165,166,167,168,169,170] or iron supplementation [51]. In low- and middle-income countries, where anemia and global micronutrient deficiency prevalence are high, the iron fortification of food can include multi-micronutrient fortification powder with demonstrated effectiveness in children under 2 years of age [181].

##### Iron Supplementation

WHO Recommendations

In areas where anemia is highly prevalent (>40%), the WHO recommends, as a public health intervention, a daily iron supplementation for 3 months of the year to all children over 6 months of age [51]. The recommended supplementations are 10–12.5 mg/day in children aged 6–23 months, 30 mg/day in children aged 24–59 months, and 30–60 mg/day in 5–12 year old children and adolescent girls [51,141]. In countries where the prevalence of anemia is 20–40%, intermittent regimens of iron supplementation can be considered.

2.Concerns about ID Prevention in Malaria-Endemic Areas

Concerns have been expressed on a possible increased risk of malaria with iron supplementation in malaria-endemic areas [6,7,51,182,183,184,185], with a reported RR of 1.16 (95% CI 1.02 to 1.31) [184]. Conversely, mild ID may protect against *Plasmodium falciparum* [6,7,182,185,186]. The mechanisms involved remain poorly understood [182,184,185]. The hypothesis raised proposes that iron-deficient erythrocytes are more resistant to *Plasmodium* invasion which, on the contrary, will target reticulocytes, whose number is increased by iron, and that the parasite needs iron for growth [182,186]. On the other hand, malaria itself contributes to the high prevalence of anemia due to increased hemolysis and significant disturbances in iron metabolism [141,185]. The subsequent inflammatory response induces an increase in hepcidin production and, hence, an inhibition of iron absorption [182,185]. As a result, <32% of aged children 0.5–5 years old are responsive to iron supplementation in malaria-endemic regions [187]. According to WHO recommendations, iron supplementation should, therefore, only be given to children who have access to strategies for the prevention, diagnosis, and treatment of malaria [51,184].

3.Adverse Effects

It is necessary to remain vigilant about the potential harmful effects, related to oxidative stress, of excessive iron dosing on growth, neurodevelopment, the gut, and the microbiome, but the literature is relatively limited and should be considered with caution [10,108,109,188,189,190,191]. For children, no tolerable upper intake level (UL) has been set for iron by the EFSA, which considers the available data insufficient to do so [3]. The UL for children was set, on the other hand, at 40 mg/day by the IOM based on a no observed adverse effect level (NOAEL) for the adverse gastrointestinal effects of 30 mg/day observed in toddlers [59]. In addition to concerns of the potential adverse effects in malaria-endemic areas discussed above, the safety of iron supplementation in children with inherited Hb disorders (mainly sickle cell disorders and thalassemia) must be clarified [187].

4.Screening for Iron Deficiency

The solution of systematic screening for ID, using invasive blood tests, in asymptomatic children is far from being recommended by official organizations, and the ferritin threshold with the best sensitivity according to age and inflammation setting is yet to be determined [188,192]. This is a common challenge facing physicians. No study has been carried out in children to demonstrate the parallelism between the biomarkers used and the reality of body iron stores, and none of the ID biochemical markers is sufficiently validated in children [10]. Moreover, the large physiological changes in iron status, erythrocytes, and Hb during early life may make biomarker interpretations more difficult [6,10]. The available selected biomarkers are the most likely to characterize the three stages in the evolution of ID (Table 2). Among them, the combined measurement of ferritin and Hb is considered the mainstay for ID diagnosis [6,10,12,19,193]. However, serum ferritin is an acute-phase protein, which may be elevated in the presence of inflammation, infection, or malignancy, limiting its usefulness for the diagnosis of ID in these settings [6,188]. A simultaneous measurement of C-reactive protein or α-1 acid glycoprotein (AGP), also known as orosomucoid, is then required to rule out or adjust the ferritin concentration [6,33,82,188,194,195]. Inflammation also triggers the expression of hepcidin, causing functional ID [1,10]. The threshold of ferritin that defines ID remains uncertain [193]. The WHO defined ID as ferritin <12 μg/L in healthy children under 5 years of age and as ferritin <15 μg/L afterward [195]. Given the physiological changes in serum ferritin concentration during the first year of life, the recently identified ferritin cutoff points were 21 and 39 μg/L at 3 months, and 11 and 23 μg/L at 6 months for boys and girls, respectively, and 10 μg/L for both genders at 9 and 12 months [196]. According to a recent meta-analysis, a ferritin threshold below 15 to 30 μg /L appears to indicate absent bone marrow iron stores in healthy adults [193]. Furthermore, a ferritin threshold <50 μg/L, corresponding to a hepcidin threshold <3 nmol/L, leading to the upregulation of iron absorption, would indicate incipient iron deficiency in young women [197]. The serum iron test may lead to a significant over- or underdiagnosis of ID and is no longer indicated by guidelines [198]. According to the EFSA, iron homeostasis should be better characterized to enable the development and validation of markers indicating adaptation to insufficient iron supply [3]. Furthermore, dose–response data should be generated for iron intake/status and functional outcomes/health endpoints, such as growth, neurodevelopment, and health outcomes, in children [3]. Few studies, reviewed by Lynch et al., have linked biomarker-specific cutoffs to neurological outcome [6]. Nevertheless, a ferritin concentration <76 μg/L in cord blood, or <35 μg/L in neonates, might be considered predictive of brain ID [6]. The selective screening of IDA might be recommended in infants and children with risk factors of IDA and in those with signs and symptoms of anemia [19,192]. In the case of IDA and in children with diagnosed ID, an underlying cause should be sought (Table 1) [12,19].

## 7. Conclusions

Iron deficiency remains a critical global health problem given the persistence of its high prevalence in many countries and the subsequent health consequences, including brain dysfunction. In young children, adequate iron stores are indeed critical for erythropoiesis and neurocognitive development. Hence, approaches aimed at reducing the risk of early life iron deficiency are of public health importance.

In infants younger than 6 months of age, prenatally acquired iron stores, along with a small amount of iron provided by breast milk, are adequate to meet the needs of most healthy full-term infants, and no additional iron is needed at least until 4 months of age, with the WHO recommendations on exclusive breastfeeding being 6 months of age. After 4 to 6 months of age, supplemental iron is needed by full-term infants, and usual dietary recommendations suggest the primary use of iron-containing solid foods and, in non-breastfed infants, the prolonged use of fortified infant formula. In toddlers, the use of fortified follow-on and then young child formula should be favored.

At any age, after the introduction of solid foods, iron-rich- or -fortified foods should be provided. Since the systemic need for iron is the major determinant of iron uptake and transfer, bioavailability is not an absolute characteristic of a food or diet per se. However, as the systemic need for iron increases, the type of diet and its influence on the bioavailability of iron become increasingly relevant. In areas where ID is highly prevalent, its prevention may require systematic iron supplementation.

Aiming to improve the effectiveness of the public health dietary prevention of ID and IDA, further efforts need to be made toward developing products at a lower cost that are readily available on a global scale, which would improve the feasibility of measures and compliance. For this purpose, biofortification seems to be a promising approach. This also calls for improved health conditions and overall nutritional status, along with an optimal partnership among public health policymakers, health actors, and the industrial sectors.

## Figures and Tables

**Figure 1 nutrients-14-01604-f001:**
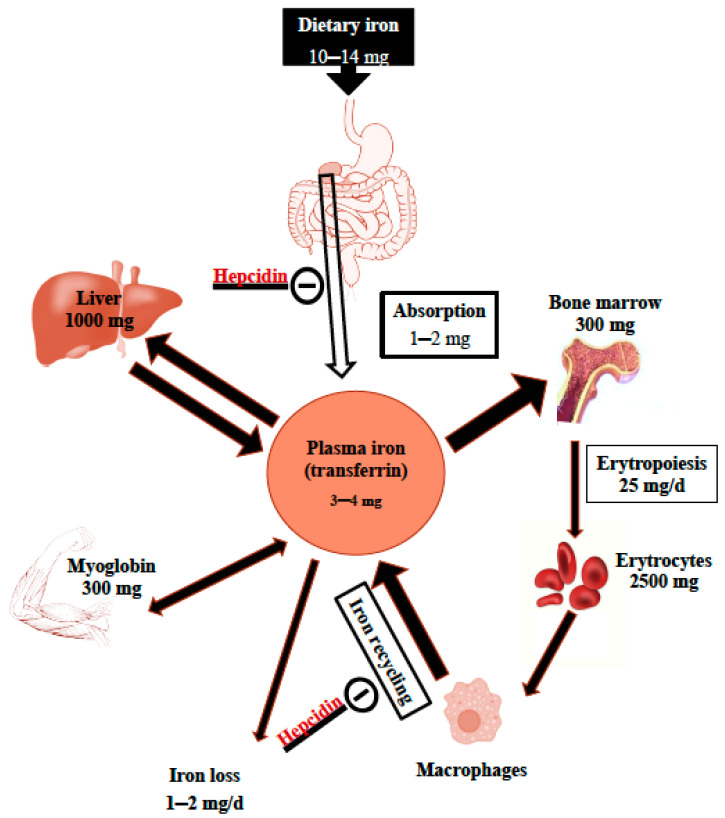
Overview of body iron distribution and daily traffic.

**Table 2 nutrients-14-01604-t002:** Stages of iron deficiency.

	Stage 1 Iron Depletion	Stage 2 Iron Deficiency Erythropoiesis without Anemia	Stage 3 Iron Deficiency Anemia
**Body iron stores**	Reduced	↓↓	↓↓↓
**Symptoms**	Nil or mild (fatigue, poor concentration)	Nil or mild (fatigue, poor concentration)	Pallor, anorexia, irritability, systolic flow murmur, tachycardia, lethargy
**Ferritin**	↓	↓↓	↓↓↓
**Transferrin saturation**	→	↓	↓
**Soluble transferrin receptors**	→	↑	↑
**Zinc protoporphyrin**	→	↑	↑
**Concentration of hemoglobin in** **reticulocytes**	→	↓	↓
**Hepcidin**	±↓	↓	↓↓
**Mean corpuscular volume**	→	→	↓
**Hemoglobin**	→	→ or ±↓	↓
**Bone marrow stainable iron**	±	0	0

↓, slight decrease; ↓↓, significant decrease; ↓↓↓, severe decrease; →, no change; ↑, increase; ±, more or less.

**Table 3 nutrients-14-01604-t003:** Overview of the most frequently used iron dietary reference values (DRVs, mg/day) set as adequate intake (AI) for 0–6 month old infants, and as the estimated average requirement (EAR) or population reference intake (PRI) for older children.

*IOM* [57]			*WHO/FAO* [59]		*EFSA* [6,58]		
Age	DRV		Age	DRV	Age	DRV	
0–6 months	**AI**: 0.27			-	0–6 months	**AI**: 0.30	
	**EAR ^a^**	**PRI**		**PRI ^b^**		**EAR ^b^**	**PRI ^b^**
7–11 months	6.9	11	7–11 months	9.3	7–11 months	8	11
1–3 years	3	7	1–3 years	5.8	1–6 years	5	7
4–8 years	4.1	10	4–6 years	6.3			

9–13 years Boys Girls	5.9 5.7	8 8	7–10 years	8.9			
					7–11 years	8	11

			11–14 years				
			Boys	14.6			
14–18 years Boys Girls	7.7 7.9	11 15	Girls	14 ^c^, 32.7	12–17 years		
			15–17 years		Boys	8	11
			Boys	18.8	Girls	7	13
			Girls	31			
			≥18 years		≥18 years		
			Boys	13.7	Boys	6	11
			Girls	29.4	Girls	7	16

^a^ Assuming a dietary iron absorption at 10% in infants 6–11 months and at 18% thereafter; ^b^ assuming a dietary iron absorption at 10%; ^c^ non-menstruating.

**Table 4 nutrients-14-01604-t004:** Overview of the main worldwide diet surveys providing the daily iron intake (DII), the percentage of children who did not meet the recommendations (dietary reference value, DRV), and the iron deficiency (ID) and iron deficiency anemia (IDA) prevalence.

Author [Reference] (Year)	Country (Study)	Age (Months, Years)	Average DII (mg/d)	Source of DRV	% DII < DRV *	ID/IDA Prevalence *
EFSA [3] (Review 2015)	Finland, France, Germany, Italy, Ireland, Latvia, Netherlands, Spain, Sweden, UK	0–11 m	2.6 to 6.0	EFSA-EAR	>50%	-/-
		12–35 m	5.0 to 7.0		<50%	
		3–10 y	7.5 to 11.5			
		10–18 y	9.2 to 14.7			
Eussen et al. [22] (Review 2015)	Albania, Austria, Belgium, Denmark, Estonia, Finland, France, Germany, Greece, Iceland, Ireland, Italy, Netherlands, Norway, Poland, Spain, Sweden, Turkey, UK	6–11 m	5.0 to 9.7	EFSA-EAR	6 to 60%	0% to 21%/-
		12–35 m	1.6 to 8.5		4 to 64%	0% to 48% (85% if breastfed)/0 to 42%
Gibson and Sidnell [63] (2014)	UK	12–17 m	6.4	British Lower Reference Nutrient Intake	13%	-/-
		18–35 m	6.4		7%	
Chouraqui et al. [64] 2020	France (Nutri-Bébé study)	0.5–5.9 m	6.4	EFSA-EAR	0%	-/-
		6–11 m	8.1		52%	
		12–35 m	7.1		30%	
ANSES [65] 2017	France (INCA 3)	0–11 m	6.6	EFSA-EAR	<50% **	-/-
		1–3 y	8.5		<50% **	
		4–6 y	7.3		>50% **	
		7–10 y	9.0		<50% **	
		11–14 y	10.1		<50% **	
		15–17 y	9.4		<50% **	
Eldridge et al. [66] (2019)	USA (FITS 2016 study)	0.5–5.9 m	7.6	-	-	-/-
		6–11.9 m	13.4			
		12–23.9 m	8.6			
		24–47.9 m	9.7			
Abrams et al. [67]	USA (daily absorbed iron calculated on the basis of data from the FITS 2016 study)	6–12 m	0.7	-	-	-/-
		Breastfed	0.3			
		Formula fed	0.9			
Atkins et al. [68] (2020)	Australia	2–5 y	7.7	IOM ***	10.1%	-/-
Harika et al. [69] (review 2017)	Ethiopia, Kenya, Nigeria, South Africa	0–6 y	3.5 to 28	WHO-EAR	13 to 100%	12% to 29%/-
Mesias et al. [58] (Review 2013)	Austria, Bolivia, Brazil, Canada, Denmark, England, Estonia, France, Germany, Greece, Hungary, Ireland, Italy, Netherlands, Norway, Perou, Scotland, Spain, Sweden, Turkey, USA	10–19 y Boys	9.0 to 24.5	-	-	-/-
		Girls	8.7 to 17.2			

* Depending on age, country, and mode of feeding; ** evaluated from the median value; *** using the full probability approach; - not provided.

**Table 5 nutrients-14-01604-t005:** Average iron content (mg/100 g of raw product as purchased minus waste) of the main iron-rich foods with range in brackets. Data mainly adapted from [114], except those indicated with reference [115]. In each category, foods are listed in descending order of iron content. The content does not presume the real contribution, which must consider the iron bioavailability and the overall composition of the meal.

Meat * and Eggs		Vegetables	
Calf’s kidney	12.0 (7.9–15)	Lentil	8.0 (5.0–13.0)
Eggs	8.8	Soya bean (dry)	6.6 (6.6–8.7)
Calf’s liver	7.9 (5.7–9.3)	Dry beans	6.5
Chicken’s liver	7.4	Chickpea	6.1 (4.9–7.2)
Black pudding	6.4 (6.4–6.5)	Topinambour ^a^	3.7 (3.4–4.0)
Sheep heart	6.1	Tofu	3.7 (2.0–5.4)
Sheep brain	3.8 (2.0-6.7)	Spinach	3.4 (1.3–7.7)
Rabbit’s meat	2.7 (1.8–6.0)	Water cress	3.1 (2.0–7.2)
Duck	2.7	Fennel	2.7
Ham	2.3 (1.7–2.9)	Lamb’s lettuce	2.0
Beef	2.1 (1.7–2.4)	Kale	1.9
Veal	2.1 (1.5–3.0)	Pea	1.6 (1.3–2.0)
Goose	1.9 (1.8–2.0)	Endive ^b^	1.4 (1.0–1.7)
Mutton	1.8 (1.5–2.7)	Mushroom	1.2 (0.7–2.0)
Pork	1.8 (0.9–2.3)	Cassava ^c^	1.2
Lamb	1.6 (1.2–1.9)	Zucchini	1.0 (0.5–2.4)
Turkey	1.0 (0.8–2.0)	Broccoli	0.8 (0.7–1.1)
Chicken	0.7 (0.6–2.0)	Leek	0.8 (0.6–1.1)
**Seafood**		**Fruits**	
Clams	7.5 [115]	Dried apricot	4.4 (3.5–5.5)
Anchovy	4.9	Dried fig	3.3 (3.0–4.0)
Mussel	4.2 (3.6–6)	Prune	2.3 (1.0–3.9)
Oyster	3.1 (2.6–7.5)	Grape (dried)	2.3
Sardine	2.4 (1.3–3.0)	Date (dried)	1.9 (1.5–2.1)
Shrimp	2.3 [115]	Green olive (marinated)	1.8 (1.6–2.0)
Herring	1.1 (0.9–1.3)	Black currant	1.3 (0.9–1.2)
Tuna	1.0	Durian ^d^	1.0 (0.8–1.1)
Salmon	0.6 (0.4–1.5)	Raspberry	1.0 (0.9–1.0)
Cod	0.3 (0.2–0.5)	Kiwi fruit	0.8 (0.3–1.6)
		Strawberry	0.7 (0.6–1.3)
**Bread and Cereals**		**Nuts**	
Wheat germ	8.6 (7.9–8.9)	Pistachio	7.3
Quinoa	8.0 (7.0–11.0)	Almond	4.1 (4.0–4.4)
Rolled oats	5.8 (4.6–6.3)	Hazelnut	3.8 (3.0–4.5)
Sorghum	5.7	Cashew nut	2.8 (1.8–3.8)
Rice (unpolished)	3.2 (2.0–3.6)	Walnut	2.5 (2.0–3.1)
Pasta made with eggs	3.0 (1.0–4.4)	Pecan nut	2.4
Wheat flours	2.2 (0.9–5.2)	Coconut	2.3 (2.0–2.7)
Whole wheat bread	2.0 (1.9–2.0)	Peanut roasted	2.3 (2.1–2.7)
Corn flakes	2.0 (1.3–2.7)	Peanut	1.8 (1.8–5.9)
Rice (polished)	0.9 (0.6–12.0)	Chestnut	1.3 (0.9–1.7)
		**Miscellaneous**	
		Honey	1.3 (0.9–2.0)
		Cane sugar (unrefined)	(1.0–8.0)
		Chocolate >40% cocoa	3.2 (2.5–4.4)
		Baker yeast	3.5 (2.1–4.9)

^a^ Topinambour is a root vegetable originating in North America which is widely cultivated across the world’s temperate zone and is relatively easy to grow. Due to its richness in inulin, it must not be introduced before the age of 3 years; ^b^ endive is a worldwide cultivated leaf vegetable with different varieties; ^c^ cassava or manioc is an important source of food in the tropics; ^d^ durian is a very popular edible fruit in Asia. *, mean and range from the different cuts of animal;

## Data Availability

Not applicable.

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
