# Peer review of "Dietary Approaches to Iron Deficiency Prevention in Childhood—A Critical Public Health Issue"

_nutrients, 2022, doi:10.3390/nu14081604_

Round 1
Reviewer 1 Report
Thank you for the opportunity to review the paper titled: "Dietary approaches to iron deficiency prevention in childhood is a critical public health issue". Although this review paper
Please note comments below.
Overall: This review is comprehensive, although it is slightly redundant in terms of the iron deficiency literature.
Introduction:
- Lines 38-43 - sentence structure needs to be reorganized
2.1 Epidemiology:
- Lines 89-91 - do you mean 'disability adjusted life years'? If so please provide relevant statistics for DALYs lost to anemia globally. https://www.karger.com/Article/PDF/345165
- Lines 90-91 -
Please update these stats with proper references - would recommend citing the Global Nutrition Report 2021 or the Lancet Series on Maternal and Child Health for more up to date statistics.
https://globalnutritionreport.org/reports/2021-global-nutrition-report/
https://www.who.int/data/gho/data/indicators/indicator-details/GHO/prevalence-of-anaemia-in-children-under-5-years-(-)
Section 3.1.1:
- Lines 136-138: What about fetal liver iron concentrations?
Section 5.2:
- Lines 261-263:
Please define which guidelines you are following for the min requirements for Americans
https://www.cmaj.ca/content/168/1/59.short
Have put a link to a paper citing Canadian guidelines
This particular section is contradictory to WHO guidelines for exclusive BF, would suggest you be as specific as possible.
https://www.who.int/news-room/fact-sheets/detail/infant-and-young-child-feeding
Lines 263-266: This point is highly context specific and needs a citation.
- Lines 263-266: It cannot be applied to LMICs contexts or needs specific clarification that it is common in Europe to wean children at 4-6 months. It is confusing given this does not have a citation based on the research literature and also does not comply with WHO recommendations to continue breastfeeding up to 2 years of age.
- Table 4: Please define uncommon food items in this list including 'topinambour, endive', which are context specific.
- needs to be defined for a lay and scientific audience - this is culturally relevant but not comprehensible to those who may not know about the region of origin/type of item it is - would suggest () or footnoting.
Other sections:
Lines 344-348 - Total diet studies at the population level may be one way to understand what community dietary practices look like to establish BCC related guidelines.
Lines 365-372 -
Consumption of tea is known to reduce iron absorption -
https://academic.oup.com/jn/article-abstract/151/9/2714/6285434
https://academic.oup.com/ajcn/article/106/6/1413/4823172?login=true
https://academic.oup.com/ajcn/article/87/4/881/4633347?login=true
Lines 452-454:
- Statement is incorrect. Please refer to links below that provide national guidelines for countries including Canada and the US.
- https://www.canada.ca/en/health-canada/services/nutrients/iron.html
- https://ods.od.nih.gov/factsheets/Iron-HealthProfessional/
- Given that your review focuses on Europe, it may be worthwhile to examine national policy guidelines for countries within the EU or EFSA guidelines and recommendations to health professionals to prevent ID using dietary approaches.
- https://www.ncbi.nlm.nih.gov/pmc/articles/PMC4179187/
Lines 499-501:
- This isn't correct - would suggest that you reference the appropriate WHO source for this recommendation.
Lines 522-525:
- Why is this the case? Can you comment on why the EFSA has not set recommendations? Does this have to do with population level iron deficiency prevalence in European populations?
- By inherited Hb disorders, are you referring to hemochromatosis? Please clarify by providing examples of what these inherited Hb disorders are - it would be ideal to also define if you are referring to iron overload/excess disorders and/or disorders causing Hb deficiency.
Lines 540-541:
- AGP and CRP are commonly used for inflammation adjustment
- https://pubmed.ncbi.nlm.nih.gov/28615259/
Please address these issues. I understand that your review is written with a European focus, but it reads as a global survey. I would encourage you to be specific about the differences in iron deficiency and anemia across high and low and middle income (LMIC) country groups, given that iron deficiency and iron deficiency anemia have differing etiologies and prevalence rates and dietary contexts will look very different for these different regions/country groups. Additionally, I would otherwise encourage you to re-focus the work to reflect only high income contexts in Europe and re-direct the reader to this so there is no confusion in terms of the context you are referring to.
Author Response
Reviewer 1:
Thank you for the opportunity to review the paper titled: "Dietary approaches to iron
deficiency prevention in childhood is a critical public health issue". Although this review
paper
Please note comments below.
Thank you very much for reviewing my paper and helping me to improve it.
First, I would like to point out to you that I was surprised to find that an error had been introduced in the order of the references in the version provided by Nutrients. This was not the case in my submissions as well in “. docx” than in “.pdf”. The error was as follows: reference 19 has been italicized and is numbered 1, which has shifted all the references that follow. I don't understand why, and apologize for it, even if I don't think it was my fault. It is not impossible that this has disturbed your review and may have made some of your comments, especially those concerning quotes, less appropriate. Anyway, I respond to all your comments below, hoping that the answers will satisfy you.
Overall: This review is comprehensive, although it is slightly redundant in terms of the iron
deficiency literature.
Thanks for this comment, I appreciate and I’m open to deletions that you would like to have the kindness to indicate to me if you consider them necessary. The manuscript has been fully reviewed by the editing service.
Introduction:
Lines 38-43 - sentence structure needs to be reorganized
Thanks for this suggestion. These sentences haves been restructured accordingly.
2.1 Epidemiology:
Lines 89-91 - do you mean 'disability adjusted life years'? If so please provide relevant
statistics for DALYs lost to anemia globally. https://www.karger.com/Article/PDF/345165
Thanks for this comment. I understand your point. However, the GBD study collaborators considered in reference 25 that “In the past decade, it has become evident that measuring non-fatal health loss is important for tracking progress as the disease burden, in terms of disability-adjusted life-years (DALYs), evolves toward being dominated by years lived with disability (YLDs)”. I nevertheless changed the wordings in order to clarify it. In the GBD study no data on global anemia were reported. The interesting study you kindly mentioned is based on data from the 2002 WHO survey. I think more appropriate to report data from 2017.
Lines 90-91 -
Please update these stats with proper references - would recommend citing the Global
Nutrition Report 2021 or the Lancet Series on Maternal and Child Health for more up to date
statistics.
https://globalnutritionreport.org/reports/2021-global-nutrition-report/
https://www.who.int/data/gho/data/indicators/indicator-details/GHO/prevalence-of-anaemiain-children-under-5-years-(-)
Thanks for this suggestion which I think is linked to an error in the citation reference (please see above in overall answer). In fact, the stats given in this sentence are those reported by the WHO in 2021. WHO reports concern the global prevalence of anemia. The subject of this review being iron deficiency, I must report the data published on this subject which are sometimes older, depending on the geographical area. The only papers I found in the Lancet that match the topic are cited as reference 25 and 27. Again because of the shift in quotes, this was surely disturbing. Very sorry for that.
Section 3.1.1:
Lines 136-138: What about fetal liver iron concentrations?
Thank you for this interesting question. Unfortunately, data on this topic in human foetus are very scarce and quite old. Anyway, I had a sentence on it with two added citations quoted as 55 and 56.
Section 5.2:
- Lines 261-263:
Please define which guidelines you are following for the min requirements for Americans
https://www.cmaj.ca/content/168/1/59.short
Have put a link to a paper citing Canadian guidelines
I'm afraid I don't understand your comment. The source of the American recommendations was quoted, namely the American Academy of Pediatrics. It became reference 75. As requested, I added the updated recommendations of Health Canada, as reference 74.
This particular section is contradictory to WHO guidelines for exclusive BF, would suggest
you be as specific as possible.
https://www.who.int/news-room/fact-sheets/detail/infant-and-young-child-feeding
Lines 263-266: This point is highly context specific and needs a citation.
Lines 263-266: It cannot be applied to LMICs contexts or needs specific clarification that it is
common in Europe to wean children at 4-6 months. It is confusing given this does not have
a citation based on the research literature and also does not comply with WHO
recommendations to continue breastfeeding up to 2 years of age.
Thanks for these comments. This section has been rewritten accordingly. It has been further developed with more details and added citations.
Table 4: Please define uncommon food items in this list including 'topinambour, endive',
which are context specific needs to be defined for a lay and scientific audience - this is culturally relevant but not comprehensible to those who may not know about the region of origin/type of item it is -would suggest () or footnoting.
Thanks for this suggestion even it’s quite difficult to know precisely which food is uncommon and for whom it is so. I think important to mention all these common foods, according to their content in iron. Anyway, I gave in footnotes some explanation for topinambour, endive, cassava and durian which perhaps are the less known in some countries.
Other sections:
Lines 344-348 - Total diet studies at the population level may be one way to understand
what community dietary practices look like to establish BCC related guidelines.
Thank you for this comment with which I totally agree even if I did not understand what BCC is. This is the case of most of the studies reported in section 4 and was indicated on the second line of this section. To go along with your comment, I clarify the sentence. It is also mentioned in section 5.1. lines 435-438 if the numbers do not change during the submission process.
Lines 365-372 -
Consumption of tea is known to reduce iron absorption -
https://academic.oup.com/jn/article-abstract/151/9/2714/6285434
https://academic.oup.com/ajcn/article/106/6/1413/4823172?login=true
https://academic.oup.com/ajcn/article/87/4/881/4633347?login=true
You are absolutely right, and it was mentioned in the penultimate line of the section 5.1, line 438, where, to be more exhaustive, I added the coffee. In the lines you mentioned, I added, after polyphenol, “such as in tea or coffee”. As the studies you kindly proposed have been performed in adults and related to iron absorption from porridge, tomato rice or wheat flour, I mentioned the most recent one when giving recommendations for adolescent in section 6.2.1.2.
Lines 452-454:
Statement is incorrect. Please refer to links below that provide national guidelines for
countries including Canada and the US.
https://www.canada.ca/en/health-canada/services/nutrients/iron.html
https://ods.od.nih.gov/factsheets/Iron-HealthProfessional/
I’m sorry but neither in the US nor in the Canadian guidelines you refer to, there are specific recommendations for children or adolescent but general recommendations that did not differ of what I wrote.
Given that your review focuses on Europe, it may be worthwhile to examine national policy
guidelines for countries within the EU or EFSA guidelines and recommendations to health
professionals to prevent ID using dietary approaches.
https://www.ncbi.nlm.nih.gov/pmc/articles/PMC4179187/
I apologize for having to disagree with your assertion. My review provides available intake data from different countries worldwide. It describes moreover specific problem linked to religion or to some parasitic infections such as malaria. It also describes some specific way of prevention which are used outside Europe, such as the use of iron cookware or ingot, as well as the importance of fortified foods and biofortification in some areas. The health policy is therefore discussed according to countries in section 6.2.2.1. Moreover, the manuscript gives the WHO recommendations of iron supplementation in areas with a high prevalence of anemia, i.e., mainly LMICs in Africa and South-East Asia.
Overall, the section has been largely rewritten to reflect your comments. I added the reference you suggested as reference 180, even if it concerns young women and not adolescents.
Lines 499-501:
This isn't correct - would suggest that you reference the appropriate WHO source for this
recommendation.
I'm afraid I don't understand your comment, because these are exactly the WHO recommendations (see below) and the correct reference n° 51.
“WHO: Recommendations1
- Daily iron supplementation is recommended as a public health intervention in infants and young
children aged 6–23 months, living in settings where anaemia is highly prevalent,2 for preventing iron
deficiency and anaemia (strong recommendation, moderate quality of evidence)….
- Daily iron supplementation is recommended as a public health intervention in preschool-age children
aged 24–59 months, living in settings where anaemia is highly prevalent,2 for increasing haemoglobin
concentrations and improving iron status (strong recommendation, very low quality of evidence).
- Daily iron supplementation is recommended as a public health intervention in school-age children
aged 60 months and older, living in settings where anaemia is highly prevalent,2 for preventing iron
deficiency and anaemia (strong recommendation, high quality of evidence).”
Lines 522-525:
Why is this the case? Can you comment on why the EFSA has not set recommendations?
Does this have to do with population level iron deficiency prevalence in European
populations?
Thanks for this comment. The sentence has been clarified accordingly both for EFSA and IOM
By inherited Hb disorders, are you referring to hemochromatosis? Please clarify by
providing examples of what these inherited Hb disorders are - it would be ideal to also
define if you are referring to iron overload/excess disorders and/or disorders causing Hb
deficiency.
As written, I do refer to inherited hemoglobin disorders such as sickle cell disease or thalassemia which lead to hemolysis and anemia. This is now clarified accordingly. Haemochromatosis which is an inherited iron overload is outside the scope of this review.
Lines 540-541:
AGP and CRP are commonly used for inflammation adjustment
https://pubmed.ncbi.nlm.nih.gov/28615259/
Thanks for this comment. AGP is also named orosomucoid, this has been clarified. The BRENDA study you mentioned was cited as reference 165 and is now 172
Please address these issues. I understand that your review is written with a European
focus, but it reads as a global survey. I would encourage you to be specific about the
differences in iron deficiency and anemia across high and low and middle income (LMIC)
country groups, given that iron deficiency and iron deficiency anemia have differing
etiologies and prevalence rates and dietary contexts will look very different for these
different regions/country groups. Additionally, I would otherwise encourage you to re-focus
the work to reflect only high income contexts in Europe and re-direct the reader to this so
there is no confusion in terms of the context you are referring to.
As above, I regret to dispute this assertion and am truly sorry that you consider this. I tried to do my best throughout this review to consider all situations and not just the situation in Western countries. Iron deficiency anemia is the consequence of severe forms of iron deficiency. Subsequently its prevalence is lower, but the etiologies are obviously the same. This is explained in section 2.2 and by table 2. I agree that anemia may have many other etiologies including other nutritional causes (Deficiencies of vitamins A, B2, B6, B12, C, D and E, folate and copper), inherited hemoglobin disorders (sickle cell diseases and thalassemia), infectious diseases. However, to include data on all these etiologies would be outside the purpose of this review which is clearly indicated in the title and in the introduction, namely iron deficiency prevention in childhood.
Reviewer 2 Report
The manuscript requires extensive editing of English language and style; in its current form it is poorly organized. Data and concepts should be presented in a more concise form.
Page 6: the paragraph “Dietary reference values for iron intake” and Table 3 are very confusing.
Data presented on page 7 (Actual dietary iron intakes in children) should be summarized in a Table
Page 9: the Authors must explain what is Formula
Figure 1: it must be indicated that the amounts of iron corresponding to absorption and iron loss refer to daily iron exchange.
Author Response
Reviewer 2
The manuscript requires extensive editing of English language and style;
Thanks for this suggestion. It has been done using the MDPI editing service with reference 42452
in its current form it is poorly organized.
The structure of the manuscript is announced at the end of the introduction. Then the article is structured in paragraphs and sub-paragraphs with each time a heading or sub-heading. I modify some sub-headings, and in the hope of better structuring the manuscript, I added some after subdividing some paragraphs in section 5 which is the main part of the manuscript.
Data and concepts should be presented in a more concise form.
I understand your point. However, it is difficult to proceed in this way at the risk of losing a great deal of information which I consider important to give to the readers, especially since reviewer 1, unless I'm mistaken, has asked for more details in certain sections
Page 6: the paragraph “Dietary reference values for iron intake” and Table 3 are very confusing.
I'm sorry, but I don't see how better to present this table. I changed its legend in hopes of making it easier to understand.
In the text I have tried to explain the reasons for the differences between the different international recommendations and gave some explanations for the readers not familiar with the concept of AI, EAR or PRI. I am totally open to all your suggestions for modifications and would be delighted to be able to give you better satisfaction by following your recommendations.
Data presented on page 7 (Actual dietary iron intakes in children) should be summarized in a Table
Thank you for this suggestion. It’s done accordingly as Table 4 giving more details especially on the countries where studies were performed.
Page 9: the Authors must explain what is Formula
This is now clarified in the section 5.2.1, page 8. Formula is the world usual used for infant and toddlers breast milk substitute according to the WHO-Codex Alimentarius definition. The corresponding citation is now provided.
Figure 1: it must be indicated that the amounts of iron corresponding to absorption and iron loss refer to daily iron exchange.
Thank you for this suggestion which I applied
Round 2
Reviewer 1 Report
Thank you for addressing my concerns and comments. This review is comprehensive and a valuable addition to the literature.
Reviewer 2 Report
The Authors answered to the main questions raised by the Reviewers and significantly improved the English language.